# Impact of sample processing delays on plasma markers of inflammation, chemotaxis, cell death, and blood coagulation

Vanessa J. Gyorffy[1,2,3], Dhruva J. Dwivedi[3,4], Patricia C. Liaw[3,4], Alison E. Fox-Robichaud[3,4], Jennifer L. Y. Tsang[2,4‡] *, Alexandra Binnie[5‡]

1 Faculty of Arts and Science, McMaster University, Hamilton, ON, Canada, 2 Niagara Health Knowledge Institute, St. Catharines, ON, Canada, 3 Thrombosis and Atherosclerosis Research Institute (TaARI), McMaster University, Hamilton, ON, Canada, 4 Department of Medicine, McMaster University, Hamilton, ON, Canada, 5 Critical Care Department, William Osler Health System, Etobicoke, ON, Canada

‡ These authors are co-senior authors.
* Jennifer.Tsang@niagarahealth.on.ca

**Data Availability Statement:** All relevant data are within the manuscript and its Supporting Information files.

## Abstract

### Background

Biosampling studies in critically ill patients traditionally involve bedside collection of samples followed by local processing (ie. centrifugation, aliquotting, and freezing) and storage. However, community hospitals, which care for the majority of Canadian patients, often lack the infrastructure for local processing and storage of specimens. A potential solution is a "simplified" biosampling protocol whereby blood samples are collected at the bedside and then shipped to a central site for processing and storage. One potential limitation of this approach is that delayed processing may alter sample characteristics.

### Objective

To determine whether delays in blood sample processing affect the stability of cytokines (IL-6, TNF, IL-10, IFN-γ), chemokines (IL-8, IP-10, MCP-1, MCP-4, MIP-1α, MIP-1β), cell-free DNA (cfDNA) (released by dying cells), and blood clotting potential in human blood samples.

### Methods

Venous blood was collected into EDTA and citrate sample tubes and stored at room temperature (RT) or 4˚C for progressive intervals up to 72 hours, prior to processing. Plasma cytokines and chemokines were quantified using single or multiplex immunoassays. cfDNA was measured using Picogreen DNA Quantification. Blood clotting potential was measured using a thrombin generation assay.

### Results

Blood samples were collected from 9 intensive care unit (ICU) patients and 7 healthy volunteers. Admission diagnoses for the ICU patients included sepsis, trauma, ruptured

**Funding:** VJG received a summer studentship from the Canadian Network of COVID-19 Clinical Trials Networks (https://www.ccctg.ca/our-initiatives/network-of-networks). No other authors received financial compensation for their work. The funder played no role in the study design, data collection or analysis, decision to publish, or preparation of the manuscript.

**Competing interests:** The authors have declared that no competing interests exist.

abdominal aortic aneurysm, intracranial hemorrhage, gastrointestinal bleed, and hyperkalemia. After pre-processing delays of up to 72 hours at RT or 4˚C, no significant changes were observed in plasma cytokines, chemokines, cfDNA, or thrombin formation.

## Conclusions

Delayed sample processing for up to 72 hours at either RT or 4˚C did not significantly affect cytokines, chemokines, cfDNA, or blood clotting potential in plasma samples from healthy volunteers and ICU patients. A "simplified" biosampling protocol is a feasible solution for conducting biosampling research at hospitals without local processing capacity.

## Introduction

Biosampling studies of plasma and serum markers in critically ill patients typically involve collection of whole blood samples at the bedside followed by immediate laboratory processing (e.g. centrifugation and aliquoting), freezing, and short-term storage at the hospital site. Community hospitals often lack the necessary infrastructure to participate in biosampling studies, in particular access to laboratory facilities for processing of research samples and short-term storage [1, 2]. As such, a "simplified" biosampling protocol in which blood samples are shipped to a central site prior to processing would facilitate the participation of community hospitals in these studies. The Genetics of Mortality for Critical Care (GenOMICC) study used this approach to collect DNA for sequencing at over 200 hospitals [3]. It is unclear, however, whether delayed processing is appropriate for plasma and serum markers that may be more sensitive to processing delays, such as cytokines, chemokines, cell free DNA (cfDNA) and blood clotting factors.

The purpose of this study was to determine whether delays in blood processing (i.e. delayed time to centrifugation, aliquotting, and freezing) of up to 72 hours affect the stability of plasma biomarkers that are commonly studied in critically ill patients [4–8]. These include cytokines (interleukin [IL]-6, IL-10, tumor necrosis factor (TNF), interferon [IFN]-γ), chemokines (IL-8, IFN-γ-induced proten 10 kDa [IP-10], monocyte chemoattractant protein [MCP]-1, MCP-4, macrophage inflammatory protein [MIP]-1α, MIP-1β), cfDNA (cell free DNA), and blood clotting potential (as assessed by thrombin generation assays). We also examined whether the type of anticoagulant (EDTA versus citrate) or the storage temperature (room temperature [RT] versus 4˚C) affected biomarker stability.

## Materials and methods

### Collection and processing of blood samples

ICU patients were recruited from the medical-surgical and neurotrauma ICUs at Hamilton General Hospital, within 24 hours of ICU admission, while healthy volunteers were recruited from the Thrombosis and Atherosclerosis Research Institute (TaARI) at McMaster University in Hamilton, ON, Canada. For each participant, blood was collected into three citrate tubes (2.7 mL each) and one EDTA tube (10 mL). The blood from the EDTA tube and the pooled citrate tubes was aliquotted into 7 Eppendorf tubes (1mL), each of which was subject to different pre-processing conditions: RT vs 4˚C, and 0, 24, 48, or 72 hours pre-processing delay. A maximum delay of 72 hours was chosen to approximate the real world situation of a sample being collected at a community hospital and shipped by courier to a central site for processing.

The RT samples were left on the lab bench in the laboratory, which was maintained at 23°C. The 4°C samples were stored in a temperature-controlled cold room (Honeywell Lab Works) with temperature confirmed by temperature tracker. After the pre-processing delay, tubes were centrifuged at 2,500 x g for 15 minutes and plasma was frozen in 60 uL aliquots at -80°C for subsequent analysis. The samples were labelled in a masked fashion to avoid bias in the assays and samples for each individual were run together in batches.

Clinical data for ICU patients was collected from the electronic medical record and included age, sex, date of ICU admission, admission diagnosis, organ supports on admission (mechanical ventilation, vasopressors, renal replacement therapy), date of blood collection, antibiotic use, blood culture results, complete blood count, lactate, and fibrinogen levels.

## Sample size calculation

Sample size calculations were performed using the G*power software (https://www.psychologie.hhu.de/arbeitsgruppen/allgemeine-psychologie-und-arbeitspsychologie/gpower). Based on a repeated measures ANOVA design with 4 timepoints, a sample size of $\geq 6$ patients or volunteers was sufficient to give $>80\%$ power to detect a medium effect size (Cohen's f) of 0.17 at an $\alpha$ error rate of 0.05.

## Quantification of IL-6 levels and cell-free plasma DNA

Plasma levels of IL-6 were quantified by enzyme-linked immunosorbent assay (ELISA) using the R&D Systems Human IL-6 Quantakine ELISA kit (Minneapolis, USA). The intra- and inter-assay coefficient of variation (CV) for the IL-6 ELISA were 2.6% and 4.5%, respectively. The IL-6 ELISA was performed using EDTA and citrate samples from ICU patients and in EDTA samples from healthy volunteers. Plasma levels of cfDNA were quantified using the Picogreen dsDNA Assay Kit (ThermoFisher Scientific, Mississauga, ON). The intra- and inter-assay CV for the cfDNA assay were 3.7% and 14%, respectively. Quality control samples ("high" and "low" samples) were included to ensure reliability.

## Multiplex analysis of cytokine and chemokine levels

Plasma levels of cytokines and chemokines in EDTA samples from ICU patients were measured using the multiplex system from Meso Scale Discovery (MSD) (Hemostasis Reference Laboratories, Hamilton, ON). Only room temperature samples were analysed. For the cytokine multiplex, the EDTA plasma samples were run "neat" (i.e. undiluted) as well as at a 1:5 dilution in MSD assay buffer. For the chemokine multiplex system, the samples were run at 1:4 and 1:10 dilutions. Two normal plasma samples were included for quality control. In a small number of ICU patient samples, the IFN-γ and TNF values were beyond the curve fit and were therefore extrapolated: these were patients 1,4,and 7 for IFN-γ and patients 1,7, and 8 for TNF. The average intra- and inter-assay CV for the multiplex cytokine assay were 3.5% and 7.6%, respectively. The average intra- and inter-assay CV for the multiplex chemokine assay were 7.9% and 5.1%, respectively.

## Thrombin generation assays

Thrombin generation assays were performed using the Technothrombin TGA reagent kit (Vienna, Austria), as previously described [9]. Briefly, citrated plasma (40 uL) was mixed with 50 uL of 1mM thrombin substrate (containing 15 mM calcium chloride [$CaCl_2$]) and coagulation was initiated by addition of Recombiplastin (relipidated tissue factor). Fluorescence was measured at 1-minute intervals over a 60-minute period at excitation and emission values of

360 and 460 nM, respectively, on a Spectramax M5e plate reader (Molecular Devices, Sunnyvale, CA, USA). Thrombin generation profiles were analyzed using TECHNOTHROMBIN TGA software (Technoclone).

## Statistical analyses

Data was analyzed using the Prism GraphPad software. Variables were expressed as mean and standard deviation or median and interquartile range (IQR), as appropriate. Changes in biomarkers were analyzed using one-way analysis of variance (ANOVA) with repeated measures.

## Ethical considerations

The study was approved by the Hamilton Integrated Research Ethics Board (HiREB #15523 and HiREB #15362). Informed written consent was obtained from all study participants, or their substitute decision makers, prior to blood collection.

## Results

We collected venous blood samples from 9 ICU patients within 24 hours of ICU admission. Baseline characteristics of the patients are shown in Table 1 and individual characteristics are shown in (S1 Table). Mean age was 61.1 ± 14.5 years and 6/9 (66.7%) were female. ICU admission diagnoses included sepsis, trauma, ruptured abdominal aortic aneurysm, intracranial hemorrhage, gastrointestinal bleed, and hyperkalemia. Venous blood was also collected from 7 healthy volunteers with a mean age of 27 ± 7 years, of whom 3/7 (42.9%) were female. The volunteers had no acute or chronic illnesses and were not taking medications at the time of enrolment.

**Table 1. Baseline characteristics of ICU patients.**

|  | ICU patients (n = 9) |
| --- | --- |
| Age (mean, years) +/- standard deviation (SD) (minimum-max) | 61.1 ± 14.5 (29–74) |
| Sex, female, N (%) | 6 (66.7%) |
| ICU admission diagnoses |  |
| Sepsis, N (%) | 2 (22.2%) |
| Trauma, N (%) | 3 (33.3%) |
| Ruptured abdominal aortic aneurysm, N (%) | 1 (11.1%) |
| Gastrointestinal bleed, N (%) | 1 (11.1%) |
| Intracranial hemorrhage, N (%) | 1 (11.1%) |
| Acute kidney injury, N (%) | 1 (11.1%) |
| SOFA (sequential organ failure assessment) score ± SD | 3.6 ± 2.7 |
| Mechanical ventilation, N (%) | 5 (56.0%) |
| Vasopressors, N (%) | 7 (77.8%) |
| Dialysis, N (%) | 1 (11.1%) |
| Platelet count (x$10^9$/L) ± SD | 149.2 ± 79.0 |
| Fibrinogen (g/L) ± SD | 2.6 ± 1.0 |
| Lactate (mM) | 4.1 ± 3.9 |
| White Blood Cell count ($10^9$/L) ± SD | 13.1 ± 4.0 |
| Use of antibiotics, N (%) | 6 (66.7%) |
| Positive blood cultures, N (%) | 3 (33.3%) |

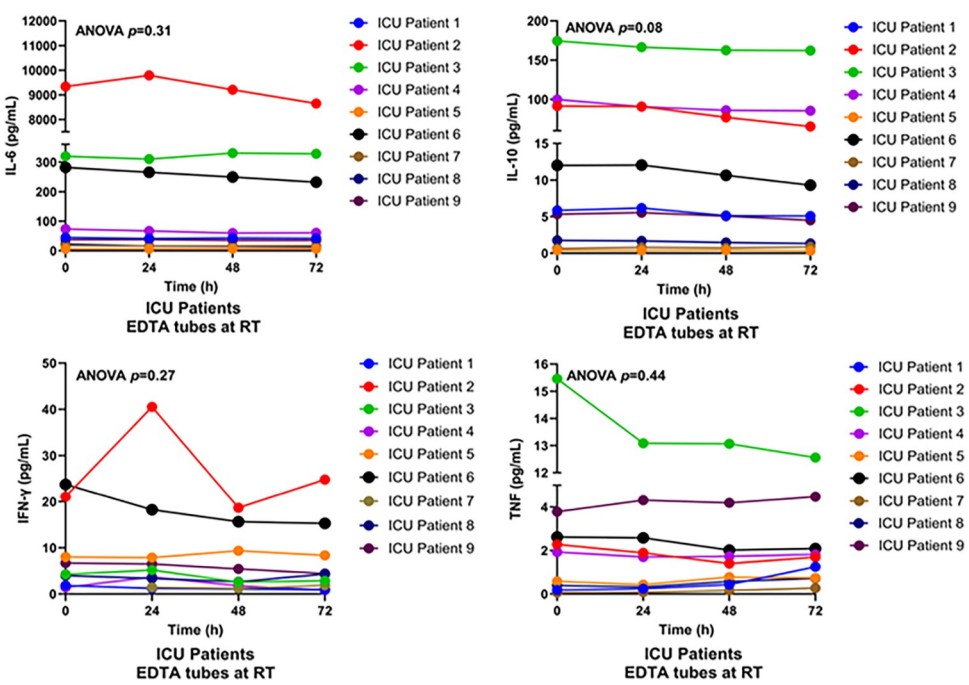

**Fig 1. Multiplex assay of cytokine levels in EDTA plasma samples from ICU patients.** Whole blood was collected from ICU patients (n = 9) into EDTA tubes and stored at room temperature (RT) for 0, 24, 48, or 72 hours prior to centrifugation. Plasma levels of IL-6, IL-10, IFN-γ, and TNF were measured using multiplex analysis.

Cytokines are soluble proteins that regulate immune and inflammatory responses [10]. To determine the impact of delayed sample processing on pro- and anti-inflammatory cytokines in critically ill patients, we used a multiplex system to measure IL-6, TNF, IL-10, and IFN-γ levels in EDTA plasma samples from ICU patients stored at RT for up to 72 hours (S2 Table). As shown in Fig 1, delayed processing did not significantly affect levels of IL-6, IL-10, TNF, or IFN-γ (ANOVA, p = 0.31, p = 0.08, p = 0.30, p = 0.32, respectively).

IL-6 is one of the most frequenty measured cytokines in critical care studies [11]. To further investigate whether the choice of anticoagulant (EDTA or citrate) and/or the storage temperature affected the stability of IL-6, we measured IL-6 levels by ELISA assay in EDTA and citrated blood samples from both ICU patients and healthy volunteers. As shown in S1 Fig, baseline IL-6 levels were significantly higher in ICU patients relative to healthy volunteers (p = 0.04). After pre-processing delays of up to 72 hours, no significant changes in IL-6 levels were observed in EDTA samples from healthy volunteers (ANOVA p = 0.67 at RT and p = 0.35 at 4°C) (S1 Fig) or ICU patients (ANOVA p = 0.55 at RT and p = 0.63 at 4°C). Similarly, no significant changes in IL-6 levels were observed in citrate samples from ICU patients at RT or 4° C (S1 Fig).

We also used a multiplex system to measure chemokines, small proteins that control the migration of immune cells in response to inflammation or infection [12]. Delayed processing of EDTA samples from ICU patients stored at RT did not significantly affect the levels of IL-8, IP-10, MCP-1, MCP-4, MIP-1α, or MIP-1β (ANOVA, p = 0.31, p = 0.21, p = 0.36, p = 0.06, p = 0.57, and p = 0.35, respectively) (Fig 2, S3 Table).

Plasma cfDNA is a marker of cell death, and is most commonly released from neutrophils as a result of NETosis [13]. After delayed processing, no significant changes in cfDNA levels were identified in either ICU patients or healthy volunteers (ANOVA p = 0.89 and p = 0.46, respectively) at RT (Fig 3) or at 4°C (S2 Fig, S4 Table).

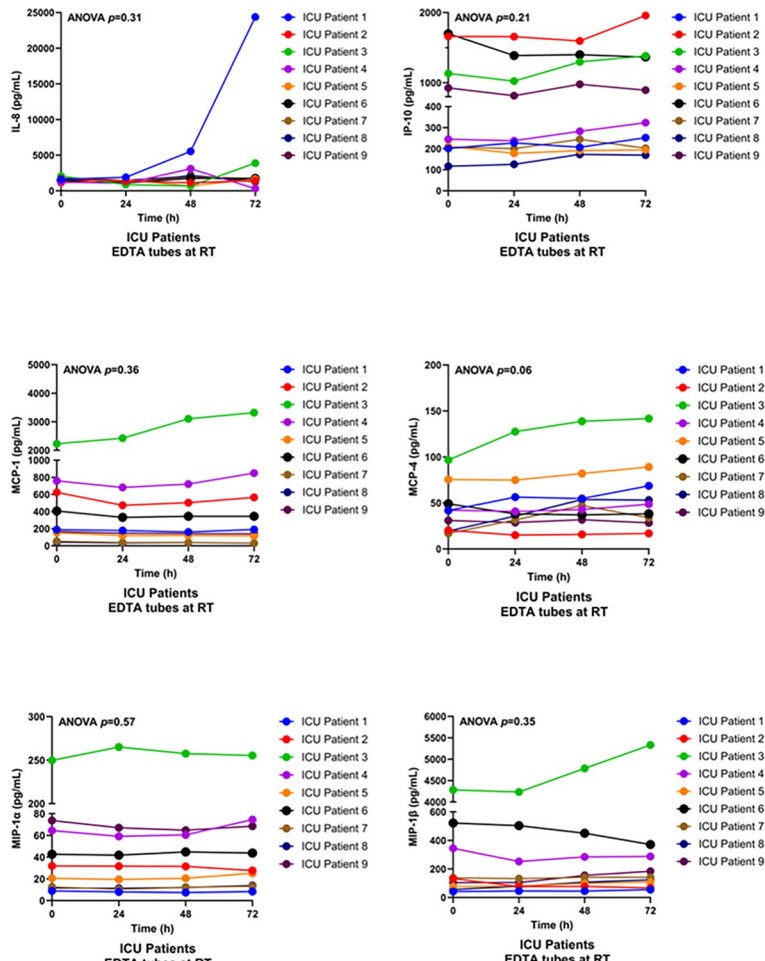

**Fig 2. Multiplex assay of chemokines levels in EDTA plasma samples from ICU patients.** Blood was collected from ICU patients (n = 9) into EDTA tubes and stored at room temperature (RT) for 0, 24, 48, or 72 hours prior to centrifugation. No significant changes were observed in levels of MCP-1, MCP-4, MIP-1α, MIP-1β, MCP-4, IP-10, or IL-8 with delayed processing at RT.

To assess coagulation potential, thrombin generation assays were used to assess thrombin formation in citrated plasma samples after pre-processing delays of 0, 24, 48 or 72 hours. This assay measures the time to initiation (expressed as lag time) as well as the total amount of thrombin formed (expressed as the area under the curve [AUC]) and can only be performed in citrated plasma samples. At baseline, the lag time was similar in healthy volunteers (3.83 ± 0.75 min) and ICU patients (6.38 ± 4.60 min) (p = 0.28), whereas the AUC was higher in healthy volunteers (4948 ± 736 versus 3492 ± 1452, p = 0.045). After delayed processing at RT, no significant changes were noted in lag time or AUC in samples from ICU patients (ANOVA, p = 0.23 and p = 0.16, respectively) or healthy volunteers (ANOVA, p = 0.60 and p = 0.11, respectively). Similarly, no significant differences in lag time or AUC were noted in samples stored at 4°C from ICU patients (ANOVA, p = 0.56 and p = 0.15, respectively), or healthy volunteers (ANOVA, p = 0.40 and p = 0.90, respectively). One ICU patient (patient 5) showed an increase in lag time and a decrease in AUC after storage at RT. However, this effect was not seen in the same patient when the sample was stored at 4°C (Fig 4, S5 Table).

## Discussion

A "simplified" biosampling approach, that does not require local sample processing, could potentially engage more community hospitals in biosampling studies. The approach entails collecting blood samples at the bedside and shipping them to a central site prior to processing and storage. However, delays in sample processing have the potential to alter assay results. In this study, we show that delays in sample processing of up to 72 hours at either RT or 4°C had no significant impact on the stability of plasma cytokines (IL-6, IL-10, TNF and IFN-γ), chemokines (IL-8, IP-10, MCP-1, MCP-4, MIP-1α, or MIP-1β), cfDNA, or blood clotting potential. Moreover, EDTA and citrated plasma samples were equally stable for cytokine, chemokine and cfDNA measurements.

Several previous studies have looked at how delays in sample processing affect plasma cytokine levels [14–16]. Thavasu et al. analysed whole blood samples (n = 5) from healthy volunteers that were spiked with recombinant cytokines. They reported that delayed processing of up to 24 hours at RT had no effect on IL-1α, IL-1β, or IFN-γ levels but modestly decreased IL-6, TNF, IFN-γ, and IFN-α levels [14]. The use of recombinant cytokines, however, may have impacted these results. Two additional studies measured endogenous cytokines in healthy volunteers and reported small changes in IL-6, TNF, IL-1β, IL-8, MIP-1α, and MIP-1β after 4 or 24 hours pre-processing delay [15, 16]. In these studies, however, plasma cytokine levels were close to the lower limit of detection of the assays, which may have impacted precision. Finally, Jackman et al. measured cytokine and chemokine levels in EDTA samples from 10 healthy volunteers and 10 trauma patients after up to 72 hours of pre-processing delay [17]. They observed no changes in TNF, IFN-γ, MIP-1α, and MIP-1β levels in healthy volunteers or trauma patients but did note an increase in IL-8 levels as well as small decreases in IL-6, IL-10, MCP-1, and IP-10 in trauma patients only. In the present study, although IL-8 levels were stable overall in both ICU patients and healthy volunteers, the one trauma patient (ICU patient #1) showed a notable 10-fold increase in IL-8 after 72 hours of pre-processing delay (Fig 2).

IL-8 is a unique chemokine that is stored in the Weibel-Palade bodies (WPBs) of vascular endothelial cells and is released upon endothelial activation or injury [18, 19]. Circulating IL-8 levels are increased in patients with trauma, sepsis, burns, and acute pancreatitis and predict the development of multiple organ failure [18]. Vascular trauma can induce the release of bone marrow-derived endothelial precursor cells (EPCs) into the circulation [19]. ICU patient #1 was the only trauma patient in the present study and also the only patient to show a significant increase in IL-8 after delayed processing. One hypothesis is that circulating endothelial cells and/or EPCs resulting from vascular trauma promote ongoing IL-8 production in trauma blood samples. Further studies will be required to confirm this hypothesis and to determine whether increases in IL-8 are consistently observed in trauma patient samples with delayed processing. Other cytokines and chemokines were not significantly altered in this patient.

Plasma cfDNA, which is released from injured or dying cells, is present at low levels in the blood of healthy individuals [20]. Elevated levels of plasma cfDNA have been reported in pathologic states including cancer, trauma, autoimmune diseases, and sepsis. In septic patients, high levels of plasma cfDNA (up to 10-fold higher than normal) are predictive of poor outcome [21, 22]. A previous study reported that cfDNA levels are stable in EDTA plasma samples for up to 6 hours at RT and 4°C [23]. In this study, we extended the pre-processing delay up to 72 hours at RT or 4°C and observed no significant changes in cfDNA levels (Fig 3). Since cfDNA can be released by circulating white blood cells, another factor that could impact plasma cfDNA levels is *ex vivo* DNA release during high-speed centrifugation. However, a previous study reported that centrifugation speeds from 400g to 16,000g had no effect on cfDNA

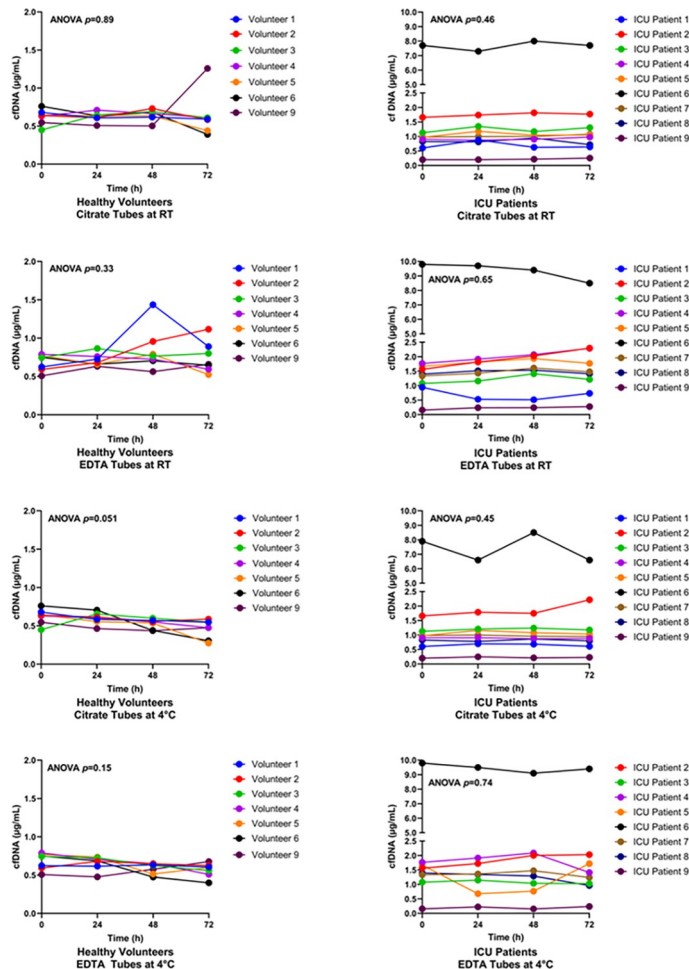

**Fig 3. cfDNA levels in citrate and EDTA plasma samples from healthy volunteers and ICU patients.** Blood was collected from healthy volunteers (n = 7) and ICU patients (n = 9) into citrate or EDTA tubes and stored at room temperature (RT) for 0, 24, 48, or 72 hours prior to centrifugation. cfDNA levels were measured using a Picogreen assay.

levels [24]. Moreover, a centralized sample processing strategy would ensure consistency in centrifugation speed, which would minimize this potential confounder.

Thrombin generation assays measure the dynamics of blood clotting, including pro- and anti-coagulant activities [25]. The assay is conducted using citrated plasma and is widely used for research purposes as well as in the diagnosis, prognosis, and treatment of bleeding and thrombotic disorders. Rapid processing of blood samples is recommended, ideally within 1 hour of blood draw [25]. In the present study, however, we found that storing citrated blood samples for up to 72 hours at RT or 4°C did not affect thrombin formation in samples from healthy volunteers or ICU patients (Fig 4). In one patient (patient 5), the lag time and AUC were more stable when the blood was stored at 4°C. Thus, cold storage of unprocessed samples at 4°C may be preferred over RT storage.

The strengths of the current study include the analysis of samples from both healthy volunteers and ICU patients (encompassing a range of biomarker concentrations), the use of multiple time points, and the inclusion of multiple types of biomarkers. With regards to limitations, although the biomarkers assessed in this study are amongst those most commonly assayed in

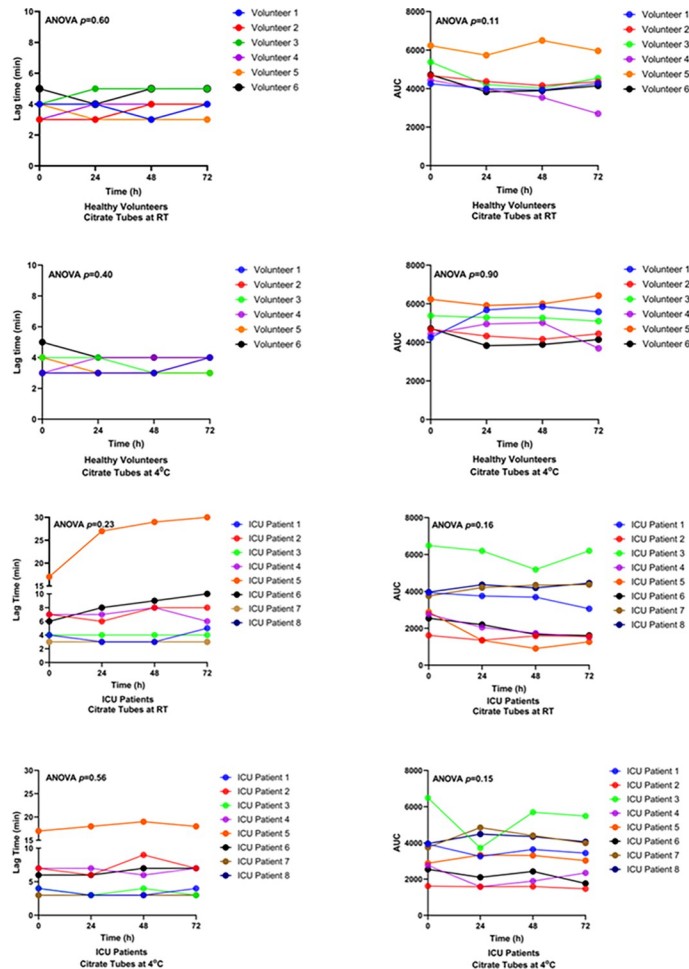

**Fig 4. Thrombin generation parameters (lag time and AUC) in citrated plasma samples from healthy volunteers and ICU patients.** Blood was collected from healthy volunteers (n = 7) and ICU patients (n = 9) into citrate tubes and stored at room temperature or at 4˚C for 0, 24, 48, or 72 hours, prior to centrifugation. Blood clotting potential was measured using a thrombin generation assay. The lag time is expressed as minutes. The units of AUC are "nM thrombin x min".

critical care studies, there are many other plasma biomarkers that were not assayed and that may not be stable with delayed processing [4, 5]. In addition, although the sample size was sufficient to exclude a medium effect size of 0.5, smaller effect sizes cannot be excluded. Finally, the marked increase in IL-8 that was observed in the only trauma patient raises the possibility of a trauma-specific effect on IL-8 that should be confirmed in a larger cohort.

## Conclusions

In summary, our results demonstrate that delays in blood processing of up to 72 hours at either RT or 4˚C did not significantly affect the stability of plasma cytokines, chemokines, cfDNA, and blood clotting factors in samples from healthy volunteers and ICU patients. Thus, a "simplified" approach to biosampling in which samples are collected at the bedside and shipped to a central site for processing may be a feasible solution to increase biosampling research participation in hospitals lacking local sample processing capacity.

## Supporting information

**S1 Fig. IL-6 levels in citrate and EDTA plasma samples from healthy volunteers and ICU patients, as measured by ELISA.** Blood was collected from healthy volunteers (n = 7) and ICU patients (n = 9) into citrate or EDTA tubes. The blood was stored at RT or 4˚C for 0, 24, 48, or 72 hours. No significant changed were observed in IL-6 levels with delayed processing conditions at RT or 4˚C.
(TIFF)

**S2 Fig.**
(TIFF)

**S1 Table. Baseline characteristics of individual ICU patients.** AAA (abdominal aortic aneurysm); AKI (acute kidney injury); GI (gastrointestinal); ICH (intracranial hemorrhage); SOFA score (sequential organ failure assessment score); N.D. (not done).
(PDF)

**S2 Table. Multiplex cytokine levels in EDTA plasma samples from ICU patients.**
(PDF)

**S3 Table. Multiplex chemokine levels in EDTA plasma samples from ICU patients.**
(PDF)

**S4 Table. Cell-free DNA levels in plasma samples from ICU patients and healthy volunteers.**
(PDF)

**S5 Table. Thrombin generation parameters in citrate plasma samples from ICU patients and healthy volunteers.**
(PDF)

## Acknowledgments

We thank Ms. Uzma Saeed for screening and recruiting ICU patients for this study and Dr. Neha Sharma and Mr. Caleb Reid for assistance with laboratory assays.

## Author Contributions

**Conceptualization:** Patricia C. Liaw, Jennifer L. Y. Tsang, Alexandra Binnie.

**Formal analysis:** Vanessa J. Gyorffy, Dhruva J. Dwivedi, Jennifer L. Y. Tsang, Alexandra Binnie.

**Investigation:** Vanessa J. Gyorffy, Dhruva J. Dwivedi, Patricia C. Liaw.

**Methodology:** Dhruva J. Dwivedi, Patricia C. Liaw, Jennifer L. Y. Tsang, Alexandra Binnie.

**Project administration:** Patricia C. Liaw, Jennifer L. Y. Tsang.

**Resources:** Alison E. Fox-Robichaud.

**Supervision:** Patricia C. Liaw, Alison E. Fox-Robichaud, Jennifer L. Y. Tsang, Alexandra Binnie.

**Writing – original draft:** Vanessa J. Gyorffy, Patricia C. Liaw, Jennifer L. Y. Tsang, Alexandra Binnie.

**Writing – review & editing:** Vanessa J. Gyorffy, Patricia C. Liaw, Alison E. Fox-Robichaud, Jennifer L. Y. Tsang, Alexandra Binnie.

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
