## [Decision Letter · Decision Letter 0]

10 Mar 2024

PONE-D-24-03943Impact of sample processing delays on plasma markers of inflammation, chemotaxis, cell death, and blood coagulationPLOS ONE

Dear Dr. Tsang,

Thank you for submitting your manuscript to PLOS ONE. After careful consideration, we feel that it has merit but does not fully meet PLOS ONE’s publication criteria as it currently stands. Therefore, we invite you to submit a revised version of the manuscript that addresses the points raised during the review process.

We appreciate the effort and work that went into your study titled "Impact of sample processing delays on plasma markers of inflammation, chemotaxis, cell death, and blood coagulation." The topic is of significant interest and relevance to the field. After careful review by two independent reviewers, we have decided that your manuscript could potentially be suitable for publication in PLOS ONE, subject to major revisions. Both reviewers acknowledge the technical soundness and the rigor of the statistical analysis of your study. However, several significant concerns have been raised that need to be addressed comprehensively in your revision.

We look forward to receiving your revised manuscript.

Kind regards,

Elvan Wiyarta, M.D.

Academic Editor

PLOS ONE

Journal Requirements:

Additional Editor Comments:

Data Completeness and Presentation:

Reviewer 2 has raised concerns about the completeness of data presentation, specifically regarding the assays for all treatment conditions. Please ensure that all relevant data, including those for control conditions and different storage times and temperatures, are fully presented and discussed. This may involve adding supplementary material if necessary.

Methodological Clarifications:

Clarify the rationale behind the 72-hour cut-off for sample processing delays. It is crucial to explain why this specific timeframe was chosen and how it aligns with the objectives of your study.

Justify the decision to measure IL-6 as the only cytokine in both citrate and EDTA tubes. A detailed explanation will help strengthen the methodology section of your paper.

Discussion and Literature Comparison:

The discussion section requires expansion to thoroughly interrogate the literature on IL-8 over time, particularly concerning patient 1 with a 10-fold increase in IL-8. Compare your findings with existing studies on IL-8 in ICU or vascular trauma contexts to provide a deeper analysis of your results.

Address the small sample size by documenting it as a Study Limitation. Discuss how the sample size may affect the generalizability of your findings and any implications for future research.

Data Availability Statement:

Address Reviewer 2's concern regarding the data availability statement. PLOS ONE requires authors to make all data underlying the findings described in their manuscript fully available without restriction. Please revise your Data Availability Statement to comply with this policy, or provide a more detailed explanation of any restrictions and how researchers can access the data if necessary.

Additional Clarifications and Information:

Provide more details on how blood samples were collected and processed, including information on maintaining two temperatures, the type and number of tubes collected from each participant, and specifics on the aliquot protocol.

Include information on the timing of assays and whether samples were labeled in a manner that prevented the laboratory staff from knowing the treatment conditions, to ensure the integrity of the data.

Reviewers' comments:

Reviewer's Responses to Questions

**Comments to the Author**

1. Is the manuscript technically sound, and do the data support the conclusions?

Reviewer #1: Yes

Reviewer #2: Partly

2. Has the statistical analysis been performed appropriately and rigorously? 

Reviewer #1: Yes

Reviewer #2: Yes

3. Have the authors made all data underlying the findings in their manuscript fully available?

Reviewer #1: Yes

Reviewer #2: No

4. Is the manuscript presented in an intelligible fashion and written in standard English?

Reviewer #1: Yes

Reviewer #2: Yes

5. Review Comments to the Author

Reviewer #1: Good study, well-written

Clarity required:

1. What determined the 72-hour cut-off?

2. Why was IL-6 the only cytokine measured in citrate and EDTA tubes?

Suggested revision:

3. The Discussion needs to interrogate the literature more on IL-8 over time eg.

Patient 1 with a 10-fold increase in IL-8. Apart from vascular trauma, is there any other explanation, or other studies on IL-8 in ICU or vascular trauma?

4. The small sample size needs to be documented as a Study Limitation.

Reviewer #2: Impact of sample processing delays on plasma markers of inflammation, chemotaxis, cell death, and blood coagulation

This manuscript describes a study of the effect of delayed processing times, temperature during delayed processing, and tube type on various biomarkers, including cytokines, chemokines, cell-free DNA, and thrombin (blood clotting potential). Samples were taken from a small number of healthy donors as well as ICU patients, and were either processed immediately or left for 24, 48, or 72 hours at either room temperature or 4oC. The authors report no effect of any of the treatment conditions on any of the biomarkers they measured, and conclude that collecting samples and shipping them to a central location for processing should be acceptable for research studies.

Major concern:

1. Not all the assays were reported for all the treatments (other than thrombin, which the authors explained could not be done on EDTA tubes). Why is this? For example, Figure 1 is cytokines from ICU patients, EDTA tubes, room temperature and Figure 2 is chemokines from ICU patients, EDTA tubes, room temperature. For both of these figures, where are the data for healthy individuals, citrate tubes, and 4 degrees C? Without seeing all the data, it is impossible to confirm that the data support the conclusions.

2. The study only included 7 healthy volunteers and 9 ICU patients. Is this enough subjects to be able to see an effect? A power calculation would be useful here.

Other comments:

3. Where were the blood samples collected from the healthy volunteers? At the same location as for the patients?

4. How many tubes total were collected from each participant? With 4 time periods and two temperatures, were there 8 EDTA and 24 citrate tubes collected? Or were there only 3 citrate and 1 EDTA tubes collected? If the latter, were the tubes aliquotted after they were collected and then put under the various time and temperature treatments? And if so, what was the aliquot protocol and what were they aliquotted into?

5. Were the exact type and number of tubes that were collected from the healthy subjects also collected from the ICU patients?

6. How were the two temperatures maintained? Were the room temperature samples left on a bench, or were they packed into a temperature controlled container? Were the 4C samples put in a refrigerator? Or packed in a cooler? If the latter, was there a temperature tracker included to confirm the samples stayed cold for the entire time? And was each set (by time-delay) packed separately?

7. When were the assays run?/how long were the samples frozen prior to running the assays?

8. Were the samples labeled in a masked fashion so that the laboratory staff did not know which sample had been treated which way?

9. Were all samples per individual run together in the same assay batch?

10. Were there any quality control samples included in the assays to ensure that the assays were running reliably?

11. Were any data collected from the healthy volunteers, such as age and sex?

12. Figure 3 is cfDNA for all treatments at room temperature, and Supplemental Figure 2 is all treatments at 4C. Why not show them together in one figure?

13. There are some extreme values for some of the biomarkers. It would be nice for the authors to put these into context. What is considered the normal range and are the extreme values clinically relevant?

14. Related to that, it is difficult to compare changes over time because some of the y-axis values are either shown with different scales, or so large that the changes cannot be seen. For example, Figure 2 IL-8, patient #1 is so extreme you can’t make out the variation for the other patients. And MCP-1, the scale is different for patient 3 compared with the other patients. Displaying the values of each biomarker at each time/temperature/tube in a table (perhaps supplemental) would be beneficial.

6. PLOS authors have the option to publish the peer review history of their article (what does this mean?). If published, this will include your full peer review and any attached files.

Reviewer #1: **Yes: **Nadine Rapiti

Reviewer #2: No

---

## [Decision Letter · Decision Letter 1]

20 May 2024

PONE-D-24-03943R1Impact of sample processing delays on plasma markers of inflammation, chemotaxis, cell death, and blood coagulationPLOS ONE

Dear Dr. Tsang,

Thank you for submitting your manuscript to PLOS ONE. After careful consideration, we feel that it has merit but does not fully meet PLOS ONE’s publication criteria as it currently stands. Therefore, we invite you to submit a revised version of the manuscript that addresses the points raised during the review process.

We look forward to receiving your revised manuscript.

Kind regards,

Elvan Wiyarta, M.D.

Academic Editor

PLOS ONE

Journal Requirements:

Reviewers' comments:

Reviewer's Responses to Questions

**Comments to the Author**

1. If the authors have adequately addressed your comments raised in a previous round of review and you feel that this manuscript is now acceptable for publication, you may indicate that here to bypass the “Comments to the Author” section, enter your conflict of interest statement in the “Confidential to Editor” section, and submit your "Accept" recommendation.

Reviewer #1: All comments have been addressed

Reviewer #2: (No Response)

2. Is the manuscript technically sound, and do the data support the conclusions?

Reviewer #1: Yes

Reviewer #2: Partly

3. Has the statistical analysis been performed appropriately and rigorously? 

Reviewer #1: Yes

Reviewer #2: Yes

4. Have the authors made all data underlying the findings in their manuscript fully available?

Reviewer #1: Yes

Reviewer #2: Yes

5. Is the manuscript presented in an intelligible fashion and written in standard English?

Reviewer #1: Yes

Reviewer #2: Yes

6. Review Comments to the Author

Reviewer #1: Corrections accepted, for choice of 72 hour time frame, IL-6 measurement in citrate and EDTA tubes and the sample size. The Discussion around the IL-8 increase improves the objectivity of the paper.

Reviewer #2: The authors have done a lot of work to address my comments. Some lingering items:

1. The authors have added additional results in supplemental tables. However, I still cannot find the following:

a. Cytokines, EDTA, ICU patients, cold storage

b. Cytokines, EDTA, healthy participants, room temp & cold storage

c. Cytokines, citrate, ICU patients room temp & cold storage

d. Cytokines, citrate, healthy participants, room temp & cold storage

e. Chemokines, EDTA, ICU patients, cold storage

f. Chemokines, EDTA, healthy participants, room temp & cold storage

g. Chemokines, citrate, ICU patients room temp & cold storage

h. Chemokines, citrate, healthy participants, room temp & cold storage

i. Also: IL-6, citrate, healthy participants is missing from Supplemental Figure 1

Were these assays run but not reported? Or not run? The methods section should specify exactly what assays were run on what conditions, and if the above assays were not run, then indicate why.

2. Figures 1, 2, and 4 seem to be missing so I was not able to compare the text and the figures (although if the data have not changed from the original version then I do not need to re-review these).

3. How many quality control samples were included in the batches, and can you report the coefficient of variation (CV) or another statistic to measure the reliability of the assays? Also, were quality control samples included in all assays, or only the IL-6 and cfDNA assays?

4. The authors indicate that they had greater than 80% power to detect a “medium effect size of 0.5”. What are the units of this 0.5? and what does “medium” mean?

7. PLOS authors have the option to publish the peer review history of their article (what does this mean?). If published, this will include your full peer review and any attached files.

Reviewer #1: **Yes: **Nadine Rapiti

Reviewer #2: No

---

## [Author Response · Author response to Decision Letter 1]

25 Aug 2024

August 19th, 2024

Dear PLOS ONE editorial team,

We thank the reviewers and the editorial team for reviewing our revised manuscript entitled “Impact of sample processing delays on plasma markers of inflammation, chemotaxis, cell death, and blood coagulation”. Our responses to the reviewers’ questions are provided below. We are submitting a revised copy of the manuscript with tracked changes as well as a clean copy. 

We thank the reviewers for their helpful feedback and comments. We hope that we have addressed their concerns appropriately.

Reviewer #1: All comments have been addressed. Corrections accepted, for choice of 72 hour time frame, IL-6 measurement in citrate and EDTA tubes and the sample size. The Discussion around the IL-8 increase improves the objectivity of the paper.

Reviewer #2: The authors have done a lot of work to address my comments. Some lingering items:

1. The authors have added additional results in supplemental tables. However, I still cannot find the following:

a. Cytokines, EDTA, ICU patients, cold storage

b. Cytokines, EDTA, healthy participants, room temp & cold storage

c. Cytokines, citrate, ICU patients room temp & cold storage

d. Cytokines, citrate, healthy participants, room temp & cold storage

e. Chemokines, EDTA, ICU patients, cold storage

f. Chemokines, EDTA, healthy participants, room temp & cold storage

g. Chemokines, citrate, ICU patients room temp & cold storage

h. Chemokines, citrate, healthy participants, room temp & cold storage

i. Also: IL-6, citrate, healthy participants is missing from Supplemental Figure 1

Were these assays run but not reported? Or not run? The methods section should specify exactly what assays were run on what conditions, and if the above assays were not run, then indicate why.

Response: Thank you for this question. Due to budget limitations, we could only perform the multiplex assay in one set of samples. We chose the ICU patient samples because we felt they were more clinically-relevant and the EDTA samples because EDTA is typically used as the anticoagulant in proteomics studies. We also chose the room temperature samples, rather than 4˚C samples, because we felt they were more likely to show a change with delayed processing and reflected the scenario in which samples are mailed without ice. We have edited the Methods section to clarify these issues. 

2. Figures 1, 2, and 4 seem to be missing so I was not able to compare the text and the figures (although if the data have not changed from the original version then I do not need to re-review these).

Response: We apologize for any confusion. The figures were included with the original submission but were left out of the resubmission as they were unchanged. We are including them again with this submission. 

3. How many quality control samples were included in the batches, and can you report the coefficient of variation (CV) or another statistic to measure the reliability of the assays? Also, were quality control samples included in all assays, or only the IL-6 and cfDNA assays?

Response: Thank you for this question. For the cfDNA and IL-6 assays, we included both “high” and “low” quality control samples. The intra- and inter-assay CV for the cfDNA assay were 3.7% and 14%, respectively. The intra- and inter-assay CV for the IL-6 ELISA were 2.6% and 4.5%, respectively. 

For the multiplex cytokine and chemokine assays, two normal plasma samples were included as quality control samples. The average intra- and inter-assay CV for the multiplex cytokine assay were 3.5% and 7.6%, respectively. The average intra- and inter-assay CV for the multiplex chemokine assay were 7.9% and 5.1%, respectively. 

For the thrombin generation assay, a normal pooled plasma sample was included in each run. The intra-assay CV for the lag time and area under the curve (AUC) were 7.9% and 5.2%, respectively. The inter-assay CV for the lag time and area under the curve (AUC) were 10.2% and 12.1 %, respectively. 

We have updated the manuscript to include the CV values for each assay.

4. The authors indicate that they had greater than 80% power to detect a “medium effect size of 0.5”. What are the units of this 0.5? and what does “medium” mean?

Response: Thank you for this question. Our original sample size calculation was based on a paired t-test design comparing an individual cytokine or chemokine in paired samples after 0 vs 72 hours of sample processing delay. Based on this calculation, a sample size of ≥ 7 patients is sufficient to give 80% power to detect a “medium” effect size (Cohen’s d) of 0.5 with an � error rate of 0.05. Cohen’s d is a unitless measure calculated from the anticipated difference between the group means divided by the standard deviation. An effect size of 0.2 is considered “small”, 0.5 is “medium”, and 0.8 is “large”. In the revised version of the manuscript, we have included the sample size calculation for a repeated measures ANOVA design. Using this experimental design, ≥6 samples is sufficient to give a Cohen’s f (effect size) of 0.17, which is considered a small to medium effect size for a repeated measures experiment. Sample size calculations were conducted using the G*power software.

---

## [Decision Letter · Decision Letter 2]

27 Sep 2024

Impact of sample processing delays on plasma markers of inflammation, chemotaxis, cell death, and blood coagulation

PONE-D-24-03943R2

Dear Dr. Tsang,

We’re pleased to inform you that your manuscript has been judged scientifically suitable for publication and will be formally accepted for publication once it meets all outstanding technical requirements.

Kind regards,

Elvan Wiyarta, M.D.

Academic Editor

PLOS ONE

Additional Editor Comments (optional):

Reviewers' comments:

Reviewer's Responses to Questions

**Comments to the Author**

1. If the authors have adequately addressed your comments raised in a previous round of review and you feel that this manuscript is now acceptable for publication, you may indicate that here to bypass the “Comments to the Author” section, enter your conflict of interest statement in the “Confidential to Editor” section, and submit your "Accept" recommendation.

Reviewer #2: All comments have been addressed

2. Is the manuscript technically sound, and do the data support the conclusions?

Reviewer #2: (No Response)

3. Has the statistical analysis been performed appropriately and rigorously? 

Reviewer #2: (No Response)

4. Have the authors made all data underlying the findings in their manuscript fully available?

Reviewer #2: (No Response)

5. Is the manuscript presented in an intelligible fashion and written in standard English?

Reviewer #2: (No Response)

6. Review Comments to the Author

Reviewer #2: (No Response)

7. PLOS authors have the option to publish the peer review history of their article (what does this mean?). If published, this will include your full peer review and any attached files.

Reviewer #2: No

---

## [Editor Report · Acceptance letter]

22 Oct 2024

PONE-D-24-03943R2 

PLOS ONE

Dear Dr. Tsang, 

I'm pleased to inform you that your manuscript has been deemed suitable for publication in PLOS ONE. Congratulations! Your manuscript is now being handed over to our production team.

Kind regards, 

on behalf of

Mr. Elvan Wiyarta 

Academic Editor

PLOS ONE